# The effects of physical exercise, parent-child interaction and peer relationship on adolescent depression: An empirical analysis based on CEPS data

**Lang Li, Kexin Ren***, **Bingbing Fan**

College of Physical Education, Jilin Normal University, Siping City, Jilin Province, China

* jlsdrkx@163.com

## Abstract

Currently, depression is the predominant mental illness impacting adolescents, causing severe damage to their overall health. Engaging in physical exercise can not only aid in restoring adolescent physical well-being but also function as a strategy to prevent depression and lower suicide rates. Drawing upon data from the China Education Panel Survey (CEPS) conducted between 2014 and 2015, this study delves into the effects of physical exercise on alleviating depressive symptoms among adolescent students and explores the underlying mechanisms through the lens of parent-child interactions and peer relationships. The mediation effect tests indicate that physical exercise can mitigate adolescent depression by reinforcing parent-child bonds and improving peer connections. Parents and educational institutions should judiciously plan the time for adolescents to engage in both academic pursuits and physical activities, and they should encourage greater participation in sports among adolescents through various means, thereby maximizing the beneficial role of physical exercise in ameliorating adolescent depression.

## Introduction

According to the World Health Organization, approximately 150 million people globally are affected by depression [1, 2]. There is an increased prevalence of depression in adolescents following puberty, and children with depression face a heightened risk of suicide [3–5]. Early interventions may involve pharmacotherapy to diminish the intensity of depressive symptoms, thereby enhancing health and preventing future recurrences of depression [6]. However, current medications have shown limited effectiveness in treating adolescent depression [7]. Physical exercise has surfaced as a promising alternative to conventional treatments for depression.

A sedentary lifestyle among adolescents represents a common challenge globally, with the overall prevalence of physical inactivity among adolescents in China reaching approximately 84.3% [8]. Enhancing physical exercise can markedly improve recovery rates from various physical illnesses, including coronary artery disease [9], hypertension [10], and diabetes [11]. Conversely, a lack of physical exercise can result in a multitude of chronic disorders [12].

**Data Availability Statement:** The minimal underlying data set necessary for replication of this study is accessible at the Open Science Framework (https://osf.io/hvx9n/) and the identifier is DOI 10.17605/OSF.IO/HVX9N.

**Funding:** The author(s) received no specific funding for this work.

**Competing interests:** The authors have declared that no competing interests exist.

Physical exercise plays a crucial role in both preventing and treating depression, with evidence supporting this claim from both biological and psychological perspectives [13]. Exercise interventions have been employed as an effective method to alleviate symptoms in children and adolescents suffering from depression [14–16]. Numerous studies have demonstrated that such interventions can diminish depressive symptoms in young individuals [17].

Parent-child interaction, which refers to the mutual activities between parents and children, is the earliest and most important external resource for children to establish. According to the ecosystem theory, continuous support from parents, as a protective factor of a good family system, favours the buffering of the negative impact of negative events on children and is a key protective factor for mental health problems such as depression in children [18]. In the case of adolescents specifically, physical activity is an important channel for enhancing parent-child interaction. There is empirical evidence that parent-child sport, as a new form of sport practice, has a positive impact on women's physical health, as well as on women's physical fitness, health perceptions and behaviours, psychological and social resilience, and the management and maintenance of family relationships [19]. However, there are some limitations to the studies mentioned above, with action programmes designed mainly for women and few studies for men.

Peer relationships are interpersonal relationships established and developed in the course of social interaction between peers or between individuals at the same level of physical and mental development, and are regarded as an effective indicator of the level of physical and mental health and social adjustment of the student body. Research has found that peer relationships provide individuals with the necessary emotional support and are effective in reducing the incidence of psychological problems such as anxiety and depression [20]. Peer relationships are indispensable in the development of adolescents and are a critical environmental factor influencing adolescent development. It has been shown that physical activity can greatly increase the number of peer groups and enhance the quality of peer relationships. For example, participation in sports can expand the friendship circle of adolescents and enable groups with similar sports interests to increase communication and exchange and build good interpersonal relationships [21].

In summary, physical activity, parent-child interaction, and peer relationships are all associated with depression. However, their joint effect on depression has been less well researched, and this study examines how physical activity affects depression and the mediating role that parent-child interactions and peer relationships play in the process. This will not only further clarify the mechanisms by which physical activity affects depression in adolescents, but will also allow the findings of this study to be used to guide practical teaching, hence the following research hypothesis:

**Hypothesis 1**: Physical exercise significantly mitigates adolescent depression.

**Hypothesis 2**: Physical exercise diminishes adolescent depression by enhancing parent-child interactions.

**Hypothesis 3**: Physical exercise alleviates adolescent depression by improving peer relationships.

## Materials and methods

### Study design and setting

**Data resource.** This study draws upon data from the China Education Panel Survey (CEPS), which was conceived and implemented by the China Center for Research and Data at

Renmin University of China. The CEPS employs a combination of sampling techniques, such as stratified, multi-stage, and probability proportional to size (PPS) methods, to distribute questionnaires among junior high school students (13–15 years old), their parents, classroom teachers, and school administrators. This approach aims to provide a thorough and systematic representation of how family, school, community, and overarching social structures influence individual educational outcomes. Given the focus of this research and the necessity for up-to-date data, the study specifically incorporates survey data from 2014–2015, encompassing 9,920 samples. The questionnaire explores numerous facets, including basic demographic details of the respondents, physical and mental health indicators, behaviors within schools and communities, and aspects of home education. After filtering out responses with critical missing values, the final valid sample comprised 9,455 participants.

## Study variables

**Outcome variable.** The dependent variable in this study was depressed mood. Participants were asked, "In the past seven days, have you felt any of the following (down, depressed, unhappy, life is not interesting, sad)?" The response options for the questionnaire were: "Never = 1, Rarely = 2, Sometimes = 3, Often = 4, Always = 5." These were recoded as "0 = never, 1 = rarely, 2 = sometimes, 3 = often, 4 = always." For computational simplicity, the coding was adjusted to "never = 0", "rarely = 0.25", "sometimes = 0.5", "often = 0.75", and "always = 1". Then, all individual indicators were summed, the mean was calculated, and finally, the result was multiplied by 100 to yield a continuous variable ranging from 0 to 100. Higher values of this variable indicate greater levels of depressed mood among adolescents. To align the dependent variable more closely with a normal distribution, a natural logarithm transformation was applied to the total depression score in the analysis.

**Key variable.** Using the question "How many days a week and minutes a day do you usually exercise" as a categorical variable to determine participation in school physical exercise, and drawing on the findings of related studies to establish criteria for participation, this study first eliminates the extreme samples of respondents who reported "more than 360 minutes of exercise each time." Then, it calculates the average daily exercise duration using the formula (average daily exercise time = number of exercise days per week × daily exercise time/7). In this study, after removing the outliers who indicated "more than 360 minutes of exercise each time," the average daily exercise duration for adolescents was calculated with the formula (average daily exercise time = number of exercise days per week × daily exercise time/7). To ensure that the independent variable aligns more closely with a normal distribution and to prevent the exclusion of samples with an average daily exercise time of zero, this study adds 0.01 to the average daily exercise times before taking the natural logarithm of the adjusted average daily exercise time. This process yields the continuous variable used to measure youth physical exercise; larger values of this variable indicate greater participation in physical activities by youth.

**Mediators.** The mediating variables selected are parent-child interaction and peer relationships, with the following measurement topics chosen.

Parent-child interaction was assessed in four dimensions: discussion and communication, daily activities, daily discipline, and study supervision, each comprising several elements. The CEPS survey included four questions on parent-child interaction: "How often do your parents discuss matters at school with you, your relationships with classmates, your relationships with teachers, and your concerns or worries?" Responses to these questions were compiled to yield a discussion interaction score ranging from 4 to 12. The survey also posed three questions regarding the likelihood of dining with parents, visiting museums, zoos, science centers, and

going out for movies, shows, games, etc., inquiring about the frequency of such daily activities in parent-child interactions. These three questions were consolidated to produce daily activities score ranging from 3 to 18. Six questions were asked about whether parents discipline their children regarding homework and tests, behavior at school, internet usage, TV time, choice of friends, and attire as part of daily discipline in parent-child interactions. The responses to these six questions were summarized to obtain a daily discipline score ranging from 6 to 18. Two questions addressed whether parents had supervised their children's studies in the past week by checking homework or providing homework guidance. The queries about study supervision in parent-child interactions were combined to generate a supervision score ranging from 2 to 8. The cumulative value of these four aspects resulted in a total parent-child interaction score ranging from 15 to 56.

Peer Relationships. Assessed through self-report by the students, this variable was measured using four questions from the CEPS: "Most of the students in my class are friendly to me"; "I consider myself easy to get along with"; "I often participate in activities organized by my school or class"; "I feel close to people in this school". A Likert 4-point scale was employed, with response options scored as 1 = completely disagree, 2 = comparatively disagree, 3 = comparatively agree, and 4 = completely agree. Higher total scores indicate better peer relationships.

**Control variables.** The study aggregated control variables across three domains: students' personal, family, and school characteristics. Individual traits include gender, whether the student is an only child, type of household, and relationship with parents; family attributes encompass the financial situation of the family, home internet access, parental relationships, father's alcohol consumption, and whether parents have confidence in the student; school features involve the nature of the school, type of region, and whether it is a boarding school. For variable definitions and descriptive statistics, please refer to Table 1.

**Table 1. Definition of variables and results of descriptive statistics.**

| Variable Name | Variable Description and Scoring Method | Sample Size | Mean Value | Standard Deviation | Minimum Value | Maximum Value |
|---|---|---|---|---|---|---|
| Depression | Sum of 5 indicators of depressed mood | 9455 | 2.293 | 0.422 | 1.609 | 3.219 |
| Physical exercise | Ln (days × time/7) | 9455 | 2.526 | 1.516 | -4.605 | 5.886 |
| Gender | 1 = male,0 = female | 8867 | 0.519 | 0.500 | 0 | 1 |
| Only-child or not | yes = 1, no = 0 | 9332 | 0.444 | 0.497 | 0 | 1 |
| Account type | Agricultural households = 1, non-agricultural households = 0 | 9122 | 0.534 | 0.499 | 0 | 1 |
| Relationship with parents | Mean of total score for relationship with mother and father | 9387 | 2.603 | 0.459 | 1 | 3 |
| Family economic situation | Difficulty—affluence 5-point scale | 9413 | 2.941 | 0.610 | 1 | 5 |
| Internet usage at home | With 2 = 2, with 1 = 1, without = 0 | 9412 | 1.387 | 0.878 | 0 | 2 |
| Relationships between parents | Good = 1, bad = 0 | 9363 | 0.899 | 0.301 | 0 | 1 |
| Father's Drinking | yes = 1, no = 0 | 9420 | 1.910 | 0.287 | 1 | 2 |
| Parental confidence in you | No confidence—Very confident: 4-point scale | 9402 | 3.069 | 0.703 | 1 | 4 |
| Nature of the school | Public = 1, private = 0 | 9455 | 0.928 | 0.258 | 0 | 1 |
| Area type | Worst-Best 5-point scale | 9256 | 3.496 | 1.635 | 1 | 5 |
| Whether boarding | yes = 1, no = 2 | 9280 | 1.697 | 0.459 | 1 | 2 |
| Parent-child interaction | Sum of 4 indicators of parent-child interaction | 8580 | 35.85 | 6.494 | 15 | 56 |
| Peer Relationships | Sum of 4 indicators of Peer relationships | 8749 | 12.24 | 2.821 | 4 | 16 |

## Statistical analysis

We first executed the basic regression analysis using the least squares method; subsequently, to cope with the problem of endogenous selection bias, we applied the propensity score matching (PSM) method; in addition, the Oster method was used to conduct a robustness test to ensure the reliability of the results. In order to delve deeper into the mechanisms underlying the association between physical activity and depression, we further employed mediation effects analyses to specifically explore the mediating roles played by parent-child interactions and peer relationships in this relationship.

All datasets were processed through state17.0 software. Two-sided P values < 0.05 were considered statistically significant.

## Result

### Influence of physical exercise on the depression of all samples using linear regression

To examine the impact of physical exercise on depression, a basic regression analysis using the least squares method was conducted. To further investigate potential influencing factors, step-wise regression was employed to present baseline regression results (refer to Table 2). In this analysis, only the core independent variable, physical exercise, is included in column (1), revealing a significant negative effect on adolescent depression at the 1% statistical level, with a coefficient estimate of -0.022(95% CI: -0.027, -0.016) for physical exercise. This means that for every unit increase in the logarithm of the average amount of physical activity, the logarithm of the level of depression decreases by an average of 0.022 units. Subsequently, control variables at the individual, family, and school levels were introduced in columns (2) to (4). The estimation outcomes indicate that physical exercise maintains a significant negative effect on adolescent depression at the 1% statistical level after accounting for individual characteristics, with a coefficient estimate for physical exercise of -0.013(95% CI: -0.019, -0.006). Upon introducing family characteristics, physical exercise continues to exhibit a significant negative effect on adolescent depression at the 5% statistical level, with a coefficient estimate of -0.008(95% CI: -0.014, -0.001). After including school characteristics, physical exercise still shows a significant negative effect on adolescent depression at the 1% statistical level, now with a coefficient

**Table 2. Physical exercise and adolescent depression baseline regression results.**

| VARIABLES | (1) | (2) | (3) | (4) |
|---|---|---|---|---|
| Physical exercise | -0.022*** | -0.013*** | -0.008** | -0.009*** |
|  | (0.003) | (0.003) | (0.003) | (0.003) |
| personal traits |  | control | control | control |
| Family characteristics |  |  | control | control |
| School characteristics |  |  |  | control |
| Constant | 2.354*** | 2.882*** | 3.319*** | 3.443*** |
|  | (0.010) | (0.028) | (0.046) | (0.052) |
| Observations | 9,455 | 8396 | 8,229 | 7,902 |
| R-squared | 0.006 | 0.065 | 0.105 | 0.109 |

Note:

*p<0.10,

**p<0.05,

***p<0.01; Robust standard errors in parentheses.

estimate of -0.009(95% CI: -0.016, -0.002). In conclusion, even after controlling for individual, family, school-level, and peer characteristic variables, physical exercise significantly and negatively affects adolescent depression, suggesting its efficacy in mitigating depressive symptoms among adolescents. This provides preliminary support for research hypothesis 1 of this paper.

## Endogenous issues

**Propensity Score Matching (PSM).**   The examination of the impact of physical exercise on adolescent depression may be subject to confounding factors that influence both variables, leading to sample selection bias and compromising the precision of the estimates. For this reason, this paper uses the propensity score matching method (PSM) to robustly test the results of the benchmark regression.

Adolescents who exercised at least three days a week for 30 minutes or more were considered to be 'regular participants' and were categorised as the treatment group. On the contrary, samples with physical exercise levels below this threshold are deemed 'physically inactive,' which is to say, the control group. This reclassification of physical exercise into a dichotomous variable of 0 or 1 meets the fundamental prerequisites for employing the PSM approach. Following the establishment of the treatment and control group samples, the PSM test was carried out in accordance with the steps outlined below. Initially, the conditional probabilities, also known as propensity scores, were computed for the treatment group samples using Logit models that were based on the pertinent control variables listed in column (4) of Table 2. Secondly, to ascertain if there are systematic variations in covariates and propensity scores between the treatment and control groups, this study performs a matching quality assessment by contrasting the kernel density function distribution alterations of the propensity score values pre-and post-matching. Fig 1 displays the alteration in propensity score values for both the treatment and control groups, represented as a function of kernel density matching, contrasting the situations before and after employing nearest neighbor matching with k = 4. As observed in Fig 1, the probability distribution propensity score values exhibited substantial the probability distributions of the two propensity score values exhibited substantial differences prior to matching, either due to inherent patterns in the sample data or the presence of unsuitable confounding variables within the control group. Following the matching process, there was a notable

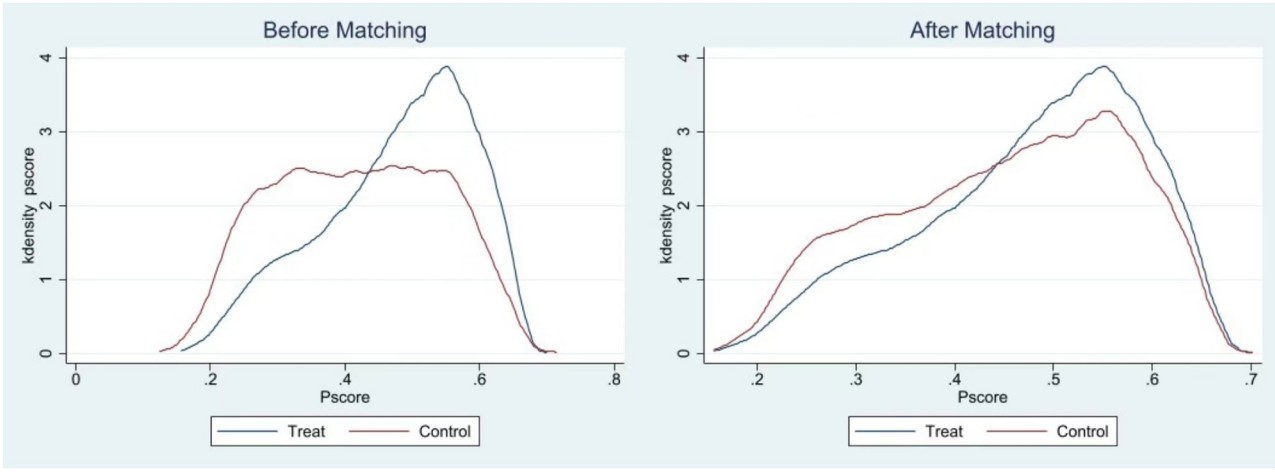

**Fig 1. Kernel density plots before and after matching.**

decrease between the kernel density equation curves there was a notable decrease in the disparity between the kernel density equation curves for the treatment and control groups, resulting in more aligned trends. The outcomes of the double t-distribution test for a single covariate and the balance test for the shift in the kernel density function distribution highlight that the employment of the PSM method diminishes the variations between the treatment and control groups concerning the explanatory variables. This aligns with the assumption of conditional independence and effectively reduces the disturbances to the precision of statistical results caused by biases in sample selection.

To precisely gauge the average treatment effect of regular physical exercise on adolescent depression, three prevalent matching techniques were employed: nearest neighbor matching (k = 4), caliper matching (r = 0.01), and kernel matching (using the default kernel function and bandwidth) for estimation purposes. As evidenced by the estimation results presented in Table 3, irrespective of the matching technique employed, the impact of regular physical exercise engagement on adolescent depression consistently exhibits a significant negative correlation. This suggests that consistent involvement in physical exercise substantially decreases the incidence of adolescent depression, with the effects varying between -0.048 and -0.040. The impact in comparison to the baseline regression outcomes, the impact, as estimated using the P the impact of physical exercise on depression, as estimated using the PSM approach, is more substantial. This augmented effect is linked to the criteria for sample grouping that are prerequisite for the application of the PSM method. Consequently, this connection elucidates, to a certain degree, the dependability of the methodology employed in this study for differentiating between the treatment and control groups. Thus, Hypothesis 1 of this paper is fully tested.

## Oster robustness test

Although this paper attempts to address endogeneity through propensity score analysis, this approach cannot control for the effects of unobserved variables. For this we use the method proposed by Oster for robustness testing [22]. Oster gives two ways to test whether omitted variables affect the empirical results: In method 1, given the ratio $\delta$ of the correlation of the omitted variable to the dependent variable to the correlation of the observable variable to the dependent variable (usually set to 1) and the maximum goodness-of-fit $R_{max}$ of the model that includes the omitted variable, the simulation yields coefficient estimates of the independent variables $\beta_1^*$; If $\beta_1^*$ lies within the 95 percent confidence interval of the $\beta_1$ estimate in the baseline regression results, then the regression results are robust. Method 2, given the model fit goodness of fit $R_{max}$ with omitted variables included and assuming that the estimated coefficients of the independent variables $\beta_1 = 0$, calculate $\delta$; if $\delta > 1$, the problem of omitted variables is not serious, and vice versa.

In this paper, we first take $R_{max} = 0.1418$ and take $R_{max}$ as 1.3 times the current regression goodness of fit and report the results of the robustness test in Table 4. The results indicate that the findings of this paper pass the robustness test.

**Table 3. PSM estimation results of physical exercise and adolescent depression.**

| Matching method | Treat | Control | ATT | Std. Dev. |
|---|---|---|---|---|
| Nearest neighbor matching | 2.253 | 2.294 | -0.040*** | 0.011 |
| caliper matching | 2.253 | 2.297 | -0.044*** | 0.010 |
| kernel matching | 2.253 | 2.302 | -0.048*** | 0.010 |

Note: The data in the table are the results obtained after 500 repeated runs using the Bootstrap method.

**Table 4. Oster robustness test.**

| Test Methods | Criteria for judgement | Actual calculation results | pass or fail |
|---|---|---|---|
| Method 1 | [-0.0149, -0.0031] | $\beta_1^* = -0.0035$ | pass |
| Method 2 | $\delta > 1$ | $\delta = 1.61925$ | pass |

## Mediation effect test

**Analysis of the mediating effects of parent-child interactions and peer relationships between physical exercise and depression.** To investigate the mechanisms governing the interplay between physical exercise and depression, this study incorporates two mediating variables: parent-child interaction and peer relationships. The objective is to analyze their mediating influences on the relationship between physical exercise and depression.

Upon establishing the mediation model, which includes control variables such as parent-child interaction, as illustrated in Fig 2, the total effect value of physical exercise on depressive symptoms was found to be -0.009($p<0.01$), while the direct effect value was -0.008($p<0.05$). The consistency in the signs of ab and c' indicates a partial mediating role of parent-child interaction in the relationship between physical exercise and depressive symptoms. The application of the same modeling approach to peer relationship mediation revealed that the direct effect value of physical exercise on depressive symptoms was -0.007($p<0.05$). Moreover, the congruence in the signs of ab and c' implies that parent-child interaction partially mediated the influence between physical exercise and depressive symptoms.

With the inclusion of control variables, parent-child and peer relationships continued to partially mediate the effects between physical activity and depressive symptoms, with indirect effects accounting for 11.5% and 19% of the total effect, respectively, and were supported by Sobel, Goodman 1, and Goodman 2 tests, illustrating the robustness of the results, as shown in Table 5. These results validate hypotheses H2 and H3.

**Bootstrap mediation effect tests for parent-child interaction, peer relationships between physical exercise and depression in adolescents.** The hypothesis testing for the mediating effects of parent-child interactions and peer relationships in the relationship between physical exercise and adolescent depression was conducted using the nonparametric percentile bootstrap test. By setting the bootstrap method to randomly sample 500 times, it was shown that none of the bootstrap confidence intervals included zero, suggesting that certain mediating effects of parent-child interactions and peer relationships in the influence of

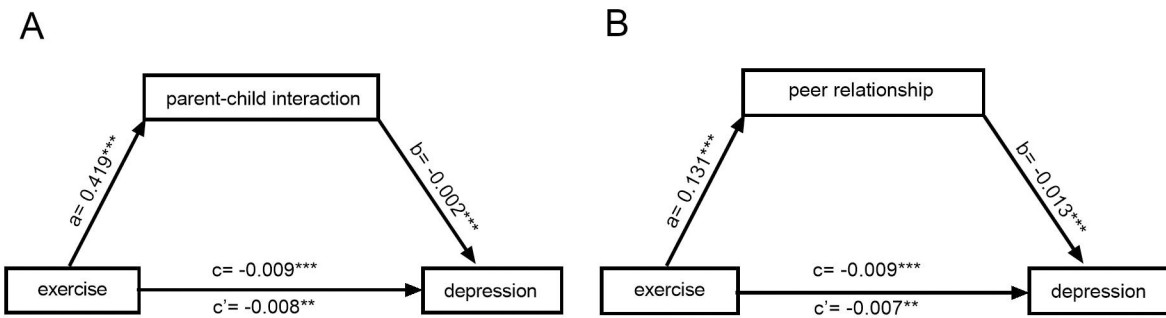

**Fig 2. The mediating effect of parent-child interaction (A) and peer relationship (B) on the impact of physical exercise on adolescent depression.**

**Table 5. Mediation effect test results.**

| VARIABLES | | Mediating variable: parent-child interaction | | | Mediating variable: peer relationship | |
|---|---|---|---|---|---|---|
| | (1) | (2) | (3) | (4) | (5) |
| | depression | parent-child interaction | depression | peer relationship | depression |
| Mediating variable | | | -0.002*** | | -0.013*** |
| | | | (0.001) | | (0.002) |
| Physical exercise | -0.009*** | 0.419*** | -0.008** | 0.131*** | -0.007** |
| | (0.003) | (0.044) | (0.003) | (0.021) | (0.003) |
| personal traits | control | control | control | control | control |
| Family characteristics | control | control | control | control | control |
| School characteristics | control | control | control | control | control |
| Constant | 3.334*** | 11.89*** | 3.398*** | 6.696*** | 3.416*** |
| | (0.043) | (0.611) | (0.046) | (0.292) | (0.045) |
| Observations | 7,902 | 7,262 | 7,262 | 7,668 | 7,668 |
| R-squared | 0.109 | 0.323 | 0.114 | 0.108 | 0.115 |
| Sobel | -0.001*** (z = -2.645) | | | -0.002*** (z = -4.858) | |
| Goodman 1 | -0.001*** (z = -2.632) | | | -0.002*** (z = -4.833) | |
| Goodman 2 | -0.001*** (z = -2.658) | | | -0.002*** (z = -4.883) | |
| Indirect effect | -0.001*** (z = -2.645) | | | -0.002*** (z = -4.858) | |
| Direct effect | -0.008** (z = -2.382) | | | -0.007** (z = -2.327) | |
| Total effect | -0.009*** (z = -2.707) | | | -0.009*** (z = -2.869) | |
| Proportion of total effect | 11.5% | | | 19.0% | |

Note: (1) Physical exercise predicts depression. (2) Physical exercise predicts parent-child interaction. (3) Physical exercise and Parent-Child Interaction Together Predict Depression. (4) Peer relationships predict depression. (5) Physical exercise and peer relationships jointly predict depression.

***p<0.01,

**p<0.05.

The parentheses indicate the t-test value.

pro-adolescent physical exercise on depression were statistically significant, as presented in Table 6.

## Discussion

The objective of this research was to explore the influence of physical exercise on depression among adolescents within the context of Chinese education, while concurrently assessing the mediating roles of parent-child interactions and peer relationships across two distinct models.

**Table 6. Bootstrap mediated effect test between adolescent physical exercise and depressive symptoms.**

| Mediating effects | Coefficient | Boot Std. Err. | 95%CI | P |
|---|---|---|---|---|
| Parent-child interactions | | | | |
| Indirect effect | -0.0010 | 0.0004 | -0.0018~-0.0002 | 0.015 |
| Direct effect | -0.0078 | 0.0035 | -0.0144~-0.0010 | 0.025 |
| Total effect | -0.0087 | 0.0035 | -0.0156~-0.0019 | 0.010 |
| Peer relationships | | | | |
| Indirect effect | -0.0017 | 0.0004 | -0.0025~-0.0009 | <0.001 |
| Direct effect | -0.0072 | 0.0034 | -0.0138~-0.0006 | 0.046 |
| Total effect | -0.0089 | 0.0033 | -0.0154~-0.0023 | 0.014 |

The findings generally upheld the research hypotheses, revealing a notable correlation between physical exercise and depression. Furthermore, both parent-child interactions and peer relationships were found to mediate the association between physical exercise and depression.

This study confirms the first hypothesis by showing that adolescents who are physically active for longer periods of time are less likely to be depressed. Earlier studies have similarly uncovered a robust connection between physical exercise and depression among adolescents, suggesting that participation in physical activities can effectively mitigate depressive symptoms in adolescents suffering from depression [23]. A meta-analysis indicates that aerobic exercises are more effective than mind-body exercises in treating adolescent depression [24]. Moreover, physical exercise has been incorporated into clinical medical practices as a tool for recovering from depression, particularly for adolescents experiencing depressive disorders [25, 26]. Exercise has been shown to modify brain structure [27], function [28], and adaptive behavior in individuals with depression [29]. For example, Woodward et al. found a significant increase in hippocampal volume after 12 weeks of exercise training in patients with schizophrenia [27]. Han Y et al. found that mood control in depressed patients could be enhanced through physical and mental exercise [28]. Oertel-Knochel et al. showed that after 4 weeks of aerobic exercise in patients with major depressive disorder, executive functioning and memory capacity improved significantly compared to pre-exercise [30]. This finding suggests that schools and families should focus on physical activity to prevent adolescent depression and promote adolescent physical and mental health through exercise interventions.

The present study confirmed the second hypothesis by establishing the partial mediating role of parent-child interaction in the relationship between physical exercise and depression among adolescents. This finding aligns with previous literature, where the Sukys study demonstrated a strong correlation between children's motivation to engage in sports and their parents' exercise habits [31]. When parents actively encourage participation in sports, there is an increase in adolescents' levels of physical exercise [32]. Furthermore, the results concerning the impact of parent-child interactions on depression are consistent with those previously reported in the literature. In a diverse sample encompassing multiple races, parental support was found to positively influence the reduction of poor mental health among adolescents, though variations existed across different racial and ethnic groups [33]. Luby's research, which focused on specific disruptive symptoms of early childhood depression, revealed that parent-child interaction therapy had a low relapse rate after 18 weeks and was as effective, if not more so, than many treatments for depression in older children [34]. During the COVID-19 pandemic lockdown, a positive association was observed between adolescent depression and the post-lockdown period, while adolescents not living with their parents were more susceptible to depression during this time [35].

The present study affirms the third hypothesis by demonstrating that peer relationships partially mediate the relationship between physical exercise and depression. Peer interactions play a crucial role in both the direct and indirect effects on intentions to adhere to exercise regimens [36, 37]. Engaging in moderate-intensity exercise habits and dedicating one to two hours to appropriate physical activities positively influence peer relationships within school environments [38]. Similarly, recreational activities aid in fostering children's social skills by enhancing their camaraderie with peers [39]. Moreover, there exists a significant negative correlation between peer relationships and depression. Peer attachment, characterized as the close bond established between adolescent individuals and their comrades, significantly impacts the psychological well-being of adolescents. International studies have indicated that peer attachment is instrumental in mitigating symptoms of adolescent depression [40]. Conversely, poor peer relationships can contribute to heightened alcohol misuse among depressed adolescents [41].

In the present study, we faced several limitations, in particular the specific challenge of identifying thresholds for categorizing physical activity participation. Furthermore, despite the initial revelation of the role of peer relationships as an important mediating variable in influencing the relationship between physical activity and mental health that we explored, we realized that there may be other mediating factors that have not yet been fully identified or understood. However, constrained by the limited resources of the current database, we were unable to uncover additional variables directly related to physical activity participation, much less capture other potential mediating variables such as academic stress. This limitation resulted in our inability to incorporate these key elements into our analytic framework, which may have limited our overall understanding of the complex relationship between physical activity and mental health. Future studies with breakthroughs in data acquisition and analytic methods will hopefully overcome these limitations and further deepen our understanding of the field.

## Conclusions

This study examined the impact of physical exercise on adolescent depression using data from the CEPS survey. The findings revealed that adolescents who participated in physical activities exhibited a significant reduction in depressive symptoms. Additionally, both parent-child interactions and peer relationships were found to play a substantial mediating roles in the influence of physical exercise on adolescent depression. Consequently, it is recommended that adolescents actively engage in physical exercise and strengthen their bonds with parents and peers through such activities as an effective strategy to prevent the onset of adolescent depression.

## Author Contributions

**Conceptualization:** Kexin Ren, Bingbing Fan.

**Formal analysis:** Lang Li, Kexin Ren, Bingbing Fan.

**Funding acquisition:** Kexin Ren.

**Investigation:** Lang Li.

**Methodology:** Lang Li, Kexin Ren, Bingbing Fan.

**Supervision:** Kexin Ren.

**Writing – original draft:** Lang Li, Kexin Ren.

**Writing – review & editing:** Kexin Ren, Bingbing Fan.

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
