## [Decision Letter · Decision Letter 0]

2 Oct 2024

PONE-D-24-41045The effects of physical exercise, parent-child interaction and peer relationship on adolescent depression: an empirical nalysis based on CEPS dataPLOS ONE

Dear Dr. Kexin,

Thank you for submitting your manuscript to PLOS ONE. After careful consideration, we feel that it has merit but does not fully meet PLOS ONE’s publication criteria as it currently stands. Therefore, we invite you to submit a revised version of the manuscript that addresses the points raised during the review process.

We look forward to receiving your revised manuscript.

Kind regards,

Henri Tilga, PhD

Academic Editor

PLOS ONE

2. Please note that your Data Availability Statement is currently missing the repository name and/or the DOI/accession number of each dataset OR a direct link to access each database. If your manuscript is accepted for publication, you will be asked to provide these details on a very short timeline. We therefore suggest that you provide this information now, though we will not hold up the peer review process if you are unable.

Reviewers' comments:

Reviewer's Responses to Questions

**Comments to the Author**

1. Is the manuscript technically sound, and do the data support the conclusions?

Reviewer #1: Partly

Reviewer #2: Yes

2. Has the statistical analysis been performed appropriately and rigorously? 

Reviewer #1: No

Reviewer #2: Yes

3. Have the authors made all data underlying the findings in their manuscript fully available?

Reviewer #1: No

Reviewer #2: Yes

4. Is the manuscript presented in an intelligible fashion and written in standard English?

Reviewer #1: Yes

Reviewer #2: Yes

5. Review Comments to the Author

Reviewer #1: Report on “The effects of physical exercise, parent-child interaction and peer relationship on adolescent depression: an empirical analysis based on CEPS data” (PONE-D-24-41045).

The paper studies the impact of physical exercise on depression using the Educational Panel Survey in China. The analysis is conducted by means of a regression analysis to find a negative correlation between exercise and depression. The authors also find that parents child interaction and peer relationship are mediators in this correlation.

The paper is well-written and presents a highly relevant research question. My main concern regards the causal interpretation of the estimation results. As a consequence of their personality, different children may have different life attitudes, and this is correlated with practicing exercise, being in a good mood and having good relationships with parents and peers. While the paper try to address the endogeneity concerns with a propensity score analysis, this approach is not valid to deal with endogeneity as it cannot control for unobserved variables (i.e. children’s personality). I suggest two alternatives to deal with this problem. One could be to apply to Oster (2019)’s test to show that the estimation results are robust to the presence of unobserved components. A second approach would be to employ instrumental variables if the authors can find valid instruments.

References

Oster, Emily. "Unobservable selection and coefficient stability: Theory and evidence." Journal of Business & Economic Statistics 37.2 (2019): 187-204.

Reviewer #2: General Overview:

The manuscript explores an important topic: adolescent depression and the roles physical exercise, parent-child interaction, and peer relationships play in mitigating its effects. Using data from the China Education Panel Survey (CEPS), the authors present a well-structured empirical analysis that examines the associations and mediating roles between these variables. The study contributes valuable insights into the prevention of adolescent depression, especially in the context of physical activity. However, there are several areas where the manuscript could be improved, particularly in terms of methodological transparency, discussion of limitations, and the clarity of the presentation.

Strengths:

Relevance of the Topic: The study addresses a critical public health issue—adolescent depression—which is highly relevant for a global audience. The emphasis on non-pharmacological interventions such as physical exercise adds a valuable perspective to mental health research.

Comprehensive Use of Data: The use of CEPS data offers a large, representative sample of adolescents, which strengthens the validity of the study’s findings. The inclusion of multiple variables (physical exercise, parent-child interaction, and peer relationships) provides a multi-dimensional approach to understanding depression in adolescents.

Clear Hypotheses: The research hypotheses are clearly stated and logically follow from the literature review. The study’s focus on both direct and mediating effects is commendable, as it allows for a more nuanced understanding of the factors affecting adolescent depression.

Weaknesses and Areas for Improvement:

Methods Section:

Transparency in Variable Selection: While the manuscript outlines the key variables and control variables used, it lacks detailed justification for some variable choices, such as the thresholds for categorizing physical exercise participation. Providing more clarity on these criteria would improve the transparency of the methodology.

Mediation Analysis: The mediation analysis is a strength of the study, but the description of the analytical approach is somewhat unclear. Expanding on the steps taken in the mediation models and explaining why certain mediators were chosen (e.g., why peer relationships were included but other potential mediators, such as academic stress, were not) would enhance the reader’s understanding.

Data Interpretation:

Presentation of Results: While the statistical results are presented clearly, there is limited interpretation of the magnitude of the effects. For example, the paper mentions that physical exercise has a significant effect on depression, but the practical significance of this finding is not fully discussed. Including more interpretation on the real-world impact of the findings (e.g., how much of a decrease in depression can be expected with increased physical activity) would be beneficial.

Effect Sizes and Confidence Intervals: To strengthen the presentation of the results, it would be helpful to include effect sizes and confidence intervals for all key findings. This would give readers a better sense of the precision and reliability of the study’s conclusions.

Discussion Section:

Limitations: The manuscript would benefit from a more thorough discussion of its limitations. For instance, the reliance on self-reported data for depression and physical exercise could introduce bias. Moreover, the study does not address potential confounders, such as socioeconomic status or academic pressure, which might also influence adolescent depression. Addressing these issues would provide a more balanced perspective on the findings.

Comparison with Existing Literature: While the discussion touches on some relevant studies, it would be helpful to more explicitly compare the findings of this study with previous research, particularly in terms of the magnitude and direction of effects. This would allow readers to better understand how this study advances the existing literature.

Figures and Tables:

The figures and tables are helpful in summarizing the results, but some could be improved. For example, the captions could provide more detailed explanations of what the figures show and how they relate to the study’s hypotheses. Additionally, ensuring that all axes in figures are clearly labeled and that the tables present sufficient detail (e.g., confidence intervals) would improve readability.

Specific Recommendations:

Enhance Clarity in Methodological Descriptions: Provide more details on the choice of thresholds for physical exercise, and clarify the steps taken in the mediation analysis.

Interpret Results with More Depth: Include more interpretation of the practical significance of the findings, and present effect sizes and confidence intervals.

Address Limitations: Discuss the potential biases introduced by self-reported data and consider additional confounders that might impact the results.

Improve Figures and Tables: Add more detail to figure captions, ensure axes are labeled clearly, and present additional statistical details (e.g., confidence intervals) in the tables.

Conclusion:

This manuscript presents important findings on the relationships between physical exercise, parent-child interaction, peer relationships, and adolescent depression. The empirical analysis is rigorous and provides valuable insights. However, the manuscript would benefit from improved clarity in methodology, a more thorough discussion of limitations, and enhanced interpretation of results. With revisions, the paper has the potential to make a strong contribution to the literature on adolescent mental health.

6. PLOS authors have the option to publish the peer review history of their article (what does this mean?). If published, this will include your full peer review and any attached files.

Reviewer #1: No

Reviewer #2: No

---

## [Author Response · Author response to Decision Letter 0]

17 Oct 2024

Dear Editor and Reviewers,

I would like to express my sincere gratitude for the opportunity to revise and resubmit our manuscript, " The effects of physical exercise, parent-child interaction and peer relationship on adolescent depression: an empirical analysis based on CEPS data", for consideration for publication in PLOS ONE. The insightful comments and suggestions from the reviewers have greatly contributed to enhancing the quality and clarity of our research. We have carefully addressed each point raised during the review process and made substantial revisions as detailed below.

Reviewer #1: The paper is well-written and presents a highly relevant research question. My main concern regards the causal interpretation of the estimation results. As a consequence of their personality, different children may have different life attitudes, and this is correlated with practicing exercise, being in a good mood and having good relationships with parents and peers.

RESPONSE: Thank you for your careful reading and positive comments on this article! Below, we respond to your comments line by line, and we welcome any further suggestions you may have.

1. While the paper tries to address the endogeneity concerns with a propensity score analysis, this approach is not valid to deal with endogeneity as it cannot control for unobserved variables (i.e. children’s personality).

RESPONSE: We have carefully considered your suggestions and taken steps to address the endogeneity concerns raised. Specifically, we conducted the Oster (2019) robustness test, as you recommended, to assess the sensitivity of our estimation results to the presence of unobserved components. This test has been added to the manuscript on page 14, line 253. The results of this analysis provide further support for our original findings, indicating that our estimates are robust to the potential influence of unobserved variables.

We believe that this additional analysis significantly strengthens our paper by providing further evidence for the causality of the relationships we investigated. We are grateful for your guidance in this matter and for suggesting such a valuable test to enhance our research.

Regarding the second approach you suggested, employing instrumental variables, we acknowledge that this could be a powerful tool to address endogeneity concerns. However, identifying valid instruments in our context is challenging and requires further research and data collection. We will continue to explore this option and consider it for future work.

Once again, thank you for your insightful comments and suggestions. We have incorporated your feedback into the manuscript and believe that it has greatly improved the quality and robustness of our research. We look forward to the next steps in the publication process.

Reviewer #2: The manuscript explores an important topic: adolescent depression and the roles physical exercise, parent-child interaction, and peer relationships play in mitigating its effects. Using data from the China Education Panel Survey (CEPS), the authors present a well-structured empirical analysis that examines the associations and mediating roles between these variables. The study contributes valuable insights into the prevention of adolescent depression, especially in the context of physical activity. However, there are several areas where the manuscript could be improved, particularly in terms of methodological transparency, discussion of limitations, and the clarity of the presentation.

RESPONSE: Thank you very much for your thorough review and constructive comments. We appreciate your efforts to improve the quality of our manuscript and have carefully addressed each of your concerns in the revised version.

1. While the manuscript outlines the key variables and control variables used, it lacks detailed justification for some variable choices, such as the thresholds for categorizing physical exercise participation. Providing more clarity on these criteria would improve the transparency of the methodology.

RENSPONSE: We deeply apologize for the lack of detailed reasons for selecting some variables in the manuscript you pointed out, especially the threshold setting for the classification of physical exercise participation. To enhance the transparency of the method, we hereby clarify and provide a detailed explanation of our method for calculating the average daily exercise time of adolescents. 

The data on physical exercise participation was sourced from the China Education Panel Survey (CEPS), which asked participants about the number of days they engaged in physical exercise per week and the average duration of each exercise session. We calculated the average daily exercise time using the formula: average daily exercise time = (weekly exercise days × average daily exercise duration / 7).

Regrettably, due to the limitations of the existing database, we were unable to classify exercise intensity based on specific thresholds. We acknowledge this as a limitation of our study and will include a discussion of the issue in the limitations section of the manuscript.

2. The mediation analysis is a strength of the study, but the description of the analytical approach is somewhat unclear. Expanding on the steps taken in the mediation models and explaining why certain mediators were chosen (e.g., why peer relationships were included but other potential mediators, such as academic stress, were not) would enhance the reader’s understanding.

RESPONE: Thank you for your valuable feedback. We fully understand your concerns about the description of intermediary analysis methods and agree that a more detailed explanation will help enhance readers' understanding. In this study, we chose peer relationships and parent-child interactions as mediating factors based on theoretical and empirical evidence (see references 18-21), as well as the availability of data. However, we acknowledge that the failure to include other potential mediating factors, such as academic pressure, may limit the comprehensiveness of our research. We discussed this limitation in detail in the discussion section and suggested that future research could consider including these factors to provide a more comprehensive perspective.

3. While the statistical results are presented clearly, there is limited interpretation of the magnitude of the effects. For example, the paper mentions that physical exercise has a significant effect on depression, but the practical significance of this finding is not fully discussed. Including more interpretation on the real-world impact of the findings (e.g., how much of a decrease in depression can be expected with increased physical activity) would be beneficial.

RESPONSE: We understand your concern about the explanation of the degree of impact and agree that providing more practical discussions will enhance the depth and breadth of the research. We discussed the practical significance of the impact of physical exercise on depression from line 321 on page 18 to line 337 on page 19. We analyzed the relationship between physical exercise and depression, and explored the potential applications of this finding in public health and clinical interventions.

We fully agree with your opinion on how much reduction in depression can be expected with increasing physical activity, which is a very good description of the results. Therefore, we have provided additional explanations in lines 183-185 of P10.

4. To strengthen the presentation of the results, it would be helpful to include effect sizes and confidence intervals for all key findings. This would give readers a better sense of the precision and reliability of the study’s conclusions.

RESPONSE: Thank you for your valuable suggestion. We fully agree on the importance of including effect size and confidence interval in enhancing the presentation of results. These statistical indicators not only provide a measure of accuracy for the results, but also enhance the accuracy and reliability of the research conclusions.

Based on your feedback, we have added a 95% confidence interval for the effect values in lines 183 to 194 of paper P10. This supplementary information helps readers understand the accuracy and credibility of statistical results. By providing confidence intervals, we can demonstrate the stability of research results and the credibility range of predicted values.

5. The manuscript would benefit from a more thorough discussion of its limitations. For instance, the reliance on self-reported data for depression and physical exercise could introduce bias. Moreover, the study does not address potential confounders, such as socioeconomic status or academic pressure, which might also influence adolescent depression. Addressing these issues would provide a more balanced perspective on the findings.

RESPONSE: We fully agree with your suggestion. Firstly, we are aware of the potential bias issues that may arise from self-reported data. To reduce the impact of selective bias, we used propensity score matching (PSM) method in our analysis. This method helps balance the observable features between the experimental group and the control group, thereby reducing bias introduced by data collection methods.

In addition, we also adopted the suggestion of another reviewer and included Oster test for robustness testing. This test helps evaluate the stability of our estimation results under different models, thereby enhancing the reliability of our conclusions.

Regarding the issue of potential confounding factors, we acknowledge that due to database limitations, we were unable to include all factors that may affect adolescent depression, such as socioeconomic status or academic pressure. We recognize that these factors may have an impact on the research results. We have already explained this limitation in the discussion section and suggest that future research consider these variables to provide a more comprehensive analysis.

6. While the discussion touches on some relevant studies, it would be helpful to more explicitly compare the findings of this study with previous research, particularly in terms of the magnitude and direction of effects. This would allow readers to better understand how this study advances the existing literature.

RESPONSE: We sincerely appreciate your suggestion, which provides us with an opportunity to further clarify the contribution of this study to existing literature. Based on your valuable feedback, we have made in-depth revisions to the discussion section.

In the revised version, we conducted a more detailed comparative analysis of the findings of this study with previous research. We paid special attention to the similarities and differences in the degree and direction of influence, and clearly pointed out how this study can fill the gaps in existing research or provide new perspectives for existing theories.

7. The figures and tables are helpful in summarizing the results, but some could be improved. For example, the captions could provide more detailed explanations of what the figures show and how they relate to the study’s hypotheses. Additionally, ensuring that all axes in figures are clearly labeled and that the tables present sufficient detail (e.g., confidence intervals) would improve readability.

RESPONSE: Regarding the captions, we have revised them to provide more detailed explanations of what each figure shows and how it relates to the study’s hypotheses. These enhancements should help readers better understand the results and their implications for our research.

Additionally, we have ensured that all axes in the figures are clearly labeled. This includes adding units where appropriate and making sure that all labels are easy to read and understand. We believe these improvements will greatly enhance the readability of the figures.

Sincerely,

Li Lang

College of Physical Education, Jilin Normal University

---

## [Decision Letter · Decision Letter 1]

25 Oct 2024

The effects of physical exercise, parent-child interaction and peer relationship on adolescent depression: an empirical analysis based on CEPS data

PONE-D-24-41045R1

Dear Dr. Kexin,

We’re pleased to inform you that your manuscript has been judged scientifically suitable for publication and will be formally accepted for publication once it meets all outstanding technical requirements.

Kind regards,

Henri Tilga, PhD

Academic Editor

PLOS ONE

Additional Editor Comments (optional):

Reviewers' comments:

Reviewer's Responses to Questions

**Comments to the Author**

1. If the authors have adequately addressed your comments raised in a previous round of review and you feel that this manuscript is now acceptable for publication, you may indicate that here to bypass the “Comments to the Author” section, enter your conflict of interest statement in the “Confidential to Editor” section, and submit your "Accept" recommendation.

Reviewer #1: All comments have been addressed

Reviewer #2: All comments have been addressed

2. Is the manuscript technically sound, and do the data support the conclusions?

Reviewer #1: Yes

Reviewer #2: (No Response)

3. Has the statistical analysis been performed appropriately and rigorously? 

Reviewer #1: Yes

Reviewer #2: (No Response)

4. Have the authors made all data underlying the findings in their manuscript fully available?

Reviewer #1: Yes

Reviewer #2: (No Response)

5. Is the manuscript presented in an intelligible fashion and written in standard English?

Reviewer #1: Yes

Reviewer #2: (No Response)

6. Review Comments to the Author

Reviewer #1: I am satisfied with the way the authors addressed my concerns. I do not put any further objection to the publication of this paper.

Reviewer #2: (No Response)

7. PLOS authors have the option to publish the peer review history of their article (what does this mean?). If published, this will include your full peer review and any attached files.

Reviewer #1: **Yes: **Juan de Dios Tena Horrillo

Reviewer #2: No

---

## [Editor Report · Acceptance letter]

5 Nov 2024

PONE-D-24-41045R1 

PLOS ONE

Dear Dr. Ren, 

I'm pleased to inform you that your manuscript has been deemed suitable for publication in PLOS ONE. Congratulations! Your manuscript is now being handed over to our production team.

Kind regards, 

on behalf of

Dr. Henri Tilga 

Academic Editor

PLOS ONE